# Synaptic counts approximate synaptic contact area in *Drosophila*

**Christopher L. Barnes**[1], **Daniel Bonnéry**[2], **Albert Cardona**[1,3]*

**1** Department of Physiology, Development and Neuroscience, University of Cambridge, Cambridge, United Kingdom, **2** Epidemiology and Modelling Group, Department of Plant Sciences, University of Cambridge, Cambridge, United Kingdom, **3** MRC Laboratory of Molecular Biology, Cambridge, United Kingdom

* acardona@mrc-lmb.cam.ac.uk

**Data Availability Statement:** The supplemental material contains all the data tables and software scripts necessary to reproduce the figures. The raw data (volume electron microscopy of a whole Drosophila larval first instar central nervous system and associated reconstructed neurons) is

## Abstract

The pattern of synaptic connections among neurons defines the circuit structure, which constrains the computations that a circuit can perform. The strength of synaptic connections is costly to measure yet important for accurate circuit modeling. Synaptic surface area has been shown to correlate with synaptic strength, yet in the emerging field of connectomics, most studies rely instead on the counts of synaptic contacts between two neurons. Here we quantified the relationship between synaptic count and synaptic area as measured from volume electron microscopy of the larval *Drosophila* central nervous system. We found that the total synaptic surface area, summed across all synaptic contacts from one presynaptic neuron to a postsynaptic one, can be accurately predicted solely from the number of synaptic contacts, for a variety of neurotransmitters. Our findings support the use of synaptic counts for approximating synaptic strength when modeling neural circuits.

## Introduction

The wiring diagram of a neuronal circuit can be represented as a directed graph whose nodes are individual neurons. Each edge represents every synaptic contact between the two nodes: the edge weight captures physiological connection strength. When reconstructed from volume electron microscopy, the edge weight is often derived from the number of synaptic contacts between two neurons: more contacts is assumed to imply a stronger connection. This approach has been used to build, on the basis of synaptic connectivity, models of neuronal circuit function. Such models are capable of predicting animal behavior with some accuracy [1–7].

The number of synaptic contacts of a neuron changes over time, due to both development and plasticity [4, 8–11]. As the neuronal arbor grows, its absolute number of synaptic contacts increases, but the fraction corresponding to specific partner neurons remains constant [4, 10]. Therefore, synaptic input fractions may be preferred over synaptic counts to compute the edge weights that approximate connection strengths, to enable comparisons across cell types and developmental stages.

However, neither the synaptic count nor the synaptic input fraction can be guaranteed to accurately predict connection strength. Two identical edge weights, as derived from either

accessible at the Virtual Fly Brain repository available online via the CATMAID web-based software at https://l1em.catmaid.virtualflybrain.org/?pid=1.

**Funding:** The work was funded by the HHMI Janelia Research Campus core funding, where Albert Cardona was a group leader between 2012 and 2019.

**Competing interests:** The authors have declared that no competing interests exist.

synaptic counts or fractions, could correspond to two different physiological connection strengths. It could be the case that some input cell types make few large synaptic contacts, and others make many smaller ones, onto the same target postsynaptic neuron. Additionally, molecular and biophysical properties of both cells and the synapse itself could invalidate both counts and fractions for deriving edge weights.

Physiologically, synaptic strength has been measured in two ways. We can derive a joint measurement for all synaptic contacts between two neurons by exciting the presynaptic neuron and recording from the postsynaptic one. Alternatively, we can count the number of vesicles that are releasing neurotransmitter at individual synaptic clefts [12–14].

The best measurement of connection strength would be paired electrophysiological recordings for all possible pairs of neurons. However, paired recordings for all possible neuron combinations is infeasible for large circuits in practice. Firstly, there is the combinatorial explosion leading to prohibitive costs. Secondly, dissected brains have limited viability *ex vivo*: only a small number of trials would be possible with any single preparation. Finally, weak connections could be hard to detect, despite being physiologically important for e.g. subthreshold potentials [15–17], or integration across many simultaneous inputs [18, 19].

One step towards evaluating synaptic strength is the quantification of synaptic surface area, which is feasible in volume electron microscopy. In mouse neocortical cells, the postsynaptic density area of a morphological synapse was shown to correlate linearly with strength of that synapse [14]. In this case, an edge weight would be a function of the total area of all synapses between two neurons, rather than simply the synapse count.

Presynaptic active zones are visible in EM primarily due to electron-dense docked vesicles and their binding machinery [20]. These vesicles represent quanta of neurotransmitter release, which cause miniature end plate potentials at the postsynaptic terminal. These quanta build up the postsynaptic potential which represents signal transduction [21]. The area of the active zone correlates strongly with the number of docked vesicles [22]. The number of docked vesicles correlates with probability of release [23, their Fig 5b], and with the strength of the synapse [24]. The total number of vesicles does not correlate with synapse strength [23, their Fig 5c]; many of these belong to the reserve pool and are only released under prolonged stimulation [12].

Similarly, postsynaptic sites are visible due to the high density of structural proteins, protein kinases and phosphates (which preferentially bind to the heavy metals used in sample preparation for electron microscopy) [25] involved in neurotransmitter reception and recycling. [26] showed that, in rat cerebellar stellate cells, the area of the postsynaptic site scaled linearly with neurotransmitter receptor count. The same work showed that variability in receptor count (and therefore postsynaptic area) is a major determinant of variability in postsynaptic current amplitude, the precursor to action potentials and therefore signal transduction. In combination, the larger the synaptic surface area is, the greater the strength of the synapse.

The cost of measuring synaptic surface areas is linear to the number of synaptic contacts for each graph edge, and could potentially be estimated by sampling only a subset of synaptic contacts. Here, we measured surface areas for all synaptic contacts of a number of connections between different cell types in the central nervous system of *Drosophila* larvae, using volume electron microscopy.

Our goal was two-fold. First, to find out whether synaptic surface areas are similar within and across cell types. Second, to study the correlation between synaptic contact number and area. We found that synaptic surface area measurements are similar across the systems and synapse types studied, and that there exists a strong correlation between the number of synaptic contacts and the total synaptic surface area per connection, which may be generalisable.

Therefore, our findings support the use of synaptic counts towards predicting connection strengths in a wiring diagram, dispensing with the labor-intensive measurements of synaptic surface areas, in *Drosophila* larvae. Therefore, any microscopy modality sufficient to resolve synaptic puncta (including light microscopy; e.g. [9, 27, 28], and more recently expansion microscopy, e.g. [29]) can serve as the basis for predicting circuit graph edge weights towards computational modeling of neuronal circuit function.

## Results

### Edge types

*Drosophila* synapses are largely one-to-many [30], and therefore the best measure of synaptic area for a single contact between two neurons is that of the postsynaptic membrane. We measured the area of 540 such contacts in 3D using a serial section transmission electron microscopy (ssTEM) image stack. These represent 80 graph edges (unique presynaptic-postsynaptic neuron partnerships). The postsynaptic neurons are first-order projection neurons belonging to two disparate sensory systems. In each system, one set of inhibitory and one set of excitatory inputs were measured bilaterally, giving 4 total edge types with left/right replication:

- Olfactory projection neurons (*PN*s) in the brain

  - Local inhibitory neurons designated "Broad D1" and "D2" (*broad*)

  - Excitatory olfactory receptor neurons (*ORN*s) onto the same pairs of olfactory PNs

- First-order mechano/nociceptive projection neurons "Basin" 1 through 4 (*Basin*s) in the first abdominal segment (a1)

  - Local inhibitory neurons designated "Drunken", "Griddle", and "Ladder" (*LN*s)

  - Stretch receptor neurons from the chordotonal organs designated "lch5", "v'ch", and "vch" (*cho*)

The neuronal arbors of all neurons had previously been reconstructed and reviewed by expert annotators [2, 31], but only as skeletonised representations, with point annotations for synaptic contacts. This approach currently represents the most feasible strategy for manual retrieval of both contact number and contact fraction. These are shown along with the neuronal morphology in Fig 1.

In the olfactory system, *broad* neurons mediate both intra- and inter-glomerular lateral inhibition [31] by synapsing onto a broad range of cell types, including ORNs and PNs. ORNs, on the other hand, primarily innervate a single partner PN (making up over 50% of its dendritic inputs in some cases), although they have some off-target edges with low contact number and fraction.

In the chordotonal system, Basin projection neurons integrate inputs from a wide array of input neurons across several modalities [2], including the mechanosensory chordotonal neurons. They mediate both short- and long-loop behavioural responses, and are repeated segmentally. Behavioural choice is in part mediated by local inhibitory neurons [3], including the LNs discussed here.

### Distribution of synaptic surface area per edge type

For every one of the 540 synaptic contacts we measured the synaptic surface area (Fig 2Ai and 2Aii; see Methods). For each edge type, we independently plotted the histograms of log-synaptic surface area for all individual morphological synapses belonging to that edge type, with

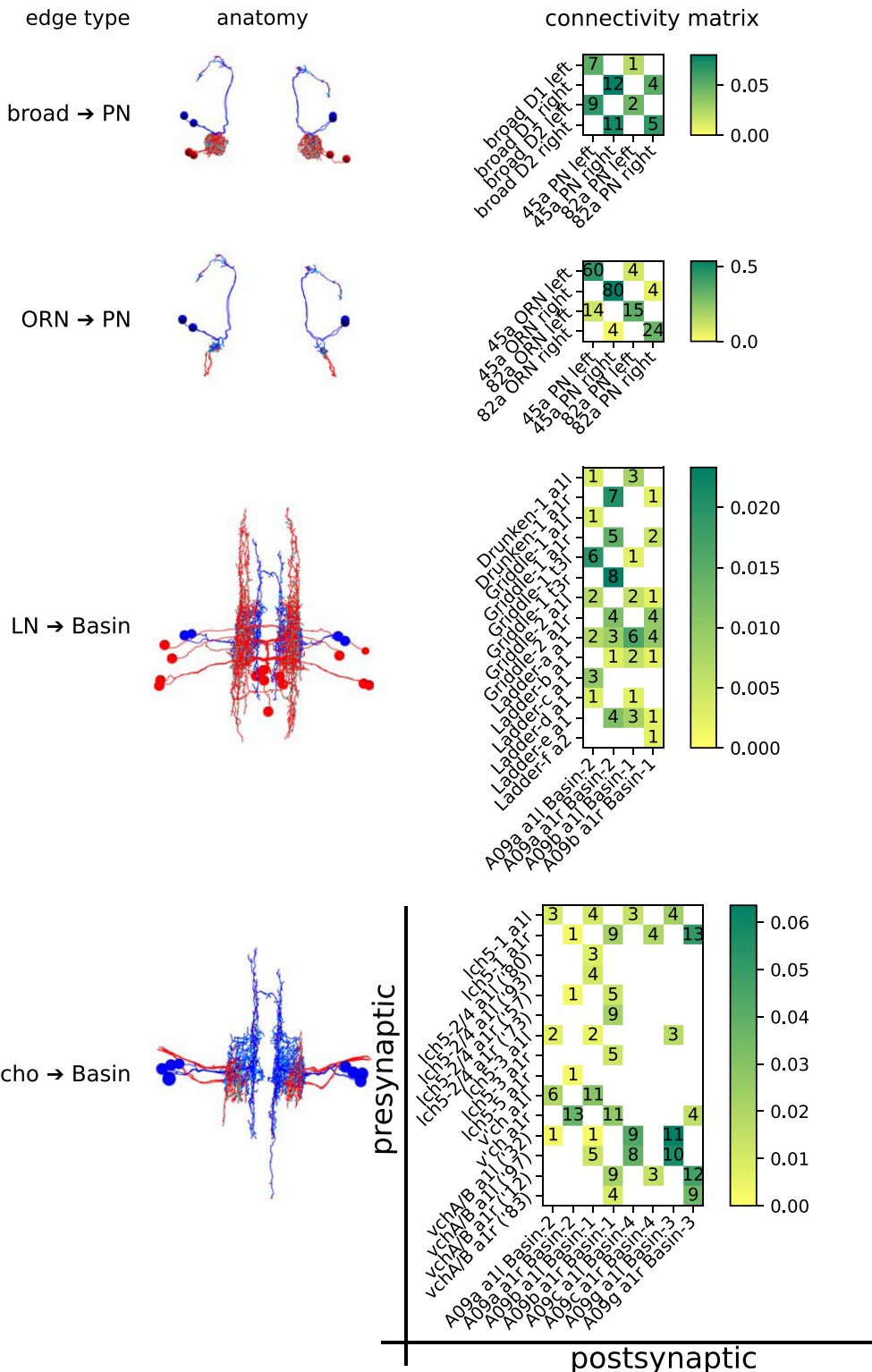

**Fig 1. The anatomy and connectivity of the 4 circuits of interest.** Presynaptic partners are shown in red, and postsynaptic in blue: synaptic sites are shown in cyan. PN-containing circuits are anterior XY projections. Basin-containing circuits are dorsal XZ projections. The connectivity matrices are have presynaptic partners on the Y axis, and postsynaptic partners on the X axis. Ambiguous pairs of neurons (lch5–2/4 and vchA/B, which are indistinguishable as their cell bodies lie outside the VNC) are distinguished by a truncated reconstruction ID. Squares in the connectivity matrix are coloured by what fraction of the target's dendritic input, by contact number, is

represented by that edge. The absolute number of contacts is also included. In total, there are 4 edge types (the 4 rows above), 80 edges (one each for each non-zero cell across all 4 matrices), and 540 contacts (each being a morphological synapse).

overlaid fitted normal distributions (Fig 2Bi and 2Bii). We chose to plot the log-surface rather than the plain surface because log-normal distributions are common in models of biological growth, where larger specimens can grow faster in absolute terms given that they have proportionally more resources than smaller specimens [32, 33].

## Correlation between synaptic input counts and synaptic surface area

To analyse whether the number of synaptic contacts can reliably predict the total synaptic surface area of an edge, we computed a linear regression between synaptic counts per edge and the sum of synaptic surface area per edge (Fig 3A). At the synaptic level, the assumption of homoscedasticity (homogeneity of variance) holds, but when aggregated at the graph edge level, the variance of the area is proportional to the count. To account for this heteroscedasticity in the aggregated model, the regression was weighted by the inverse of the synaptic count. When considering all edges jointly, ignoring edge type, we found an $R^2$ of 0.905 (Fig 3C), with visually few outliers, and a very good intercept at zero.

To analyse the influence of the edge type, we independently computed weighted linear regression for all edges of each edge type (Fig 3B). We found that all independent regressions presented similarly high $R^2$ values, indicating that correlation is strong within each edge type as well as for the joint.

While the joint regression appears very similar to each independent regression by edge type, there could be significant differences by edge type that would prevent generalisation across edge types. We then analysed the contribution of edge type to predicting total synaptic area per edge from the count of synaptic contacts per edge (the contact number per edge).

## Variability across edge types

If across edge types, an edge's total synaptic area correlated with its synaptic count, we could generalise and predict with confidence the former from the latter, independently of edge type. If not, and edge type instead had some predictive power over the synapse size, then every edge type's synaptic areas would need to be sampled before predicting total synaptic area for any edge of that type. If the predictive power of edge type is small, then synapses of different edge types can be treated as belonging to the same distribution: edge type could be ignored. Therefore, we could confidently predict total surface area for an edge independently of the edge type.

Fig 2 shows a large variance in the areas of individual synapses (the $\alpha = 0.1$ confidence interval spans approximately an order of magnitude within each edge type). The difference in means between the different edge types appears to have a small effect size in comparison to the intra-type variation (Fig 2Bi), even where that difference is, strictly speaking, statistically significant (Fig 2Biii). However, when inferring contact area from contact number in a graph edge, the edge type may still have predictive power. We here examine whether this is the case.

Bayesian hierarchical modelling allows us to incorporate our understanding of the sources of variation (e.g. potentially the edge type) into the model of a stochastic process.

Each graph edge from presynaptic neuron $i$ to postsynaptic neuron $j$ has $n$ contacts. Each contact $k$ in the edge has an area $A_{(i,j,k)}$; therefore each edge has a total area $A_{(i,j)} = \sum_{k=1}^{n_{(i,j)}} A_{(i,j,k)}$.

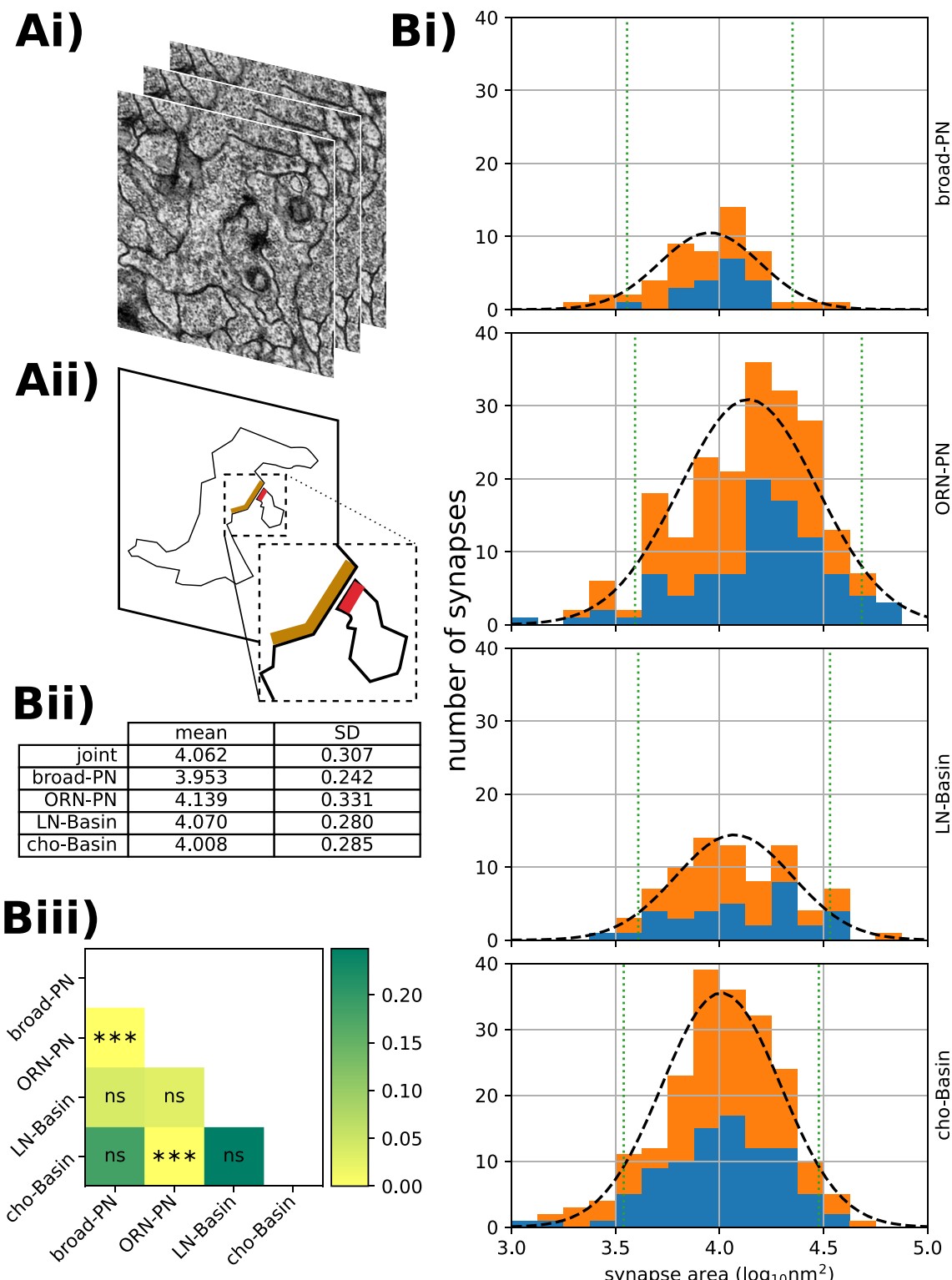

**Fig 2. Results of synaptic area labelling. Ai)** The ssTEM images upon which the analysis is based. **Aii)** A schematic showing the plane of a neurite sectioned in the top image, and the pre- and post-synaptic sites of that neurite and one of its partners, in orange and red respectively. Note the T-bar, cleft and and postsynaptic membrane specialisation. **Bi)** For each of the four circuits, the distribution of synaptic areas on a $log_{10}$ scale. The number of synapses targeting a left-sided neuron are shown in blue; right-sided in orange. Each is overlaid with the best-fitting normal distribution (black dashed line) and 90% confidence interval (green dotted line). **Bii)** Table of normal distribution parameters, in $log_{10}$ $nm^2$ to 3 decimal places. **Biii)** Raw *p*-values (colouring) and FWER corrected [34] significance levels for pairwise ranksum comparisons of circuit synapse area distributions.

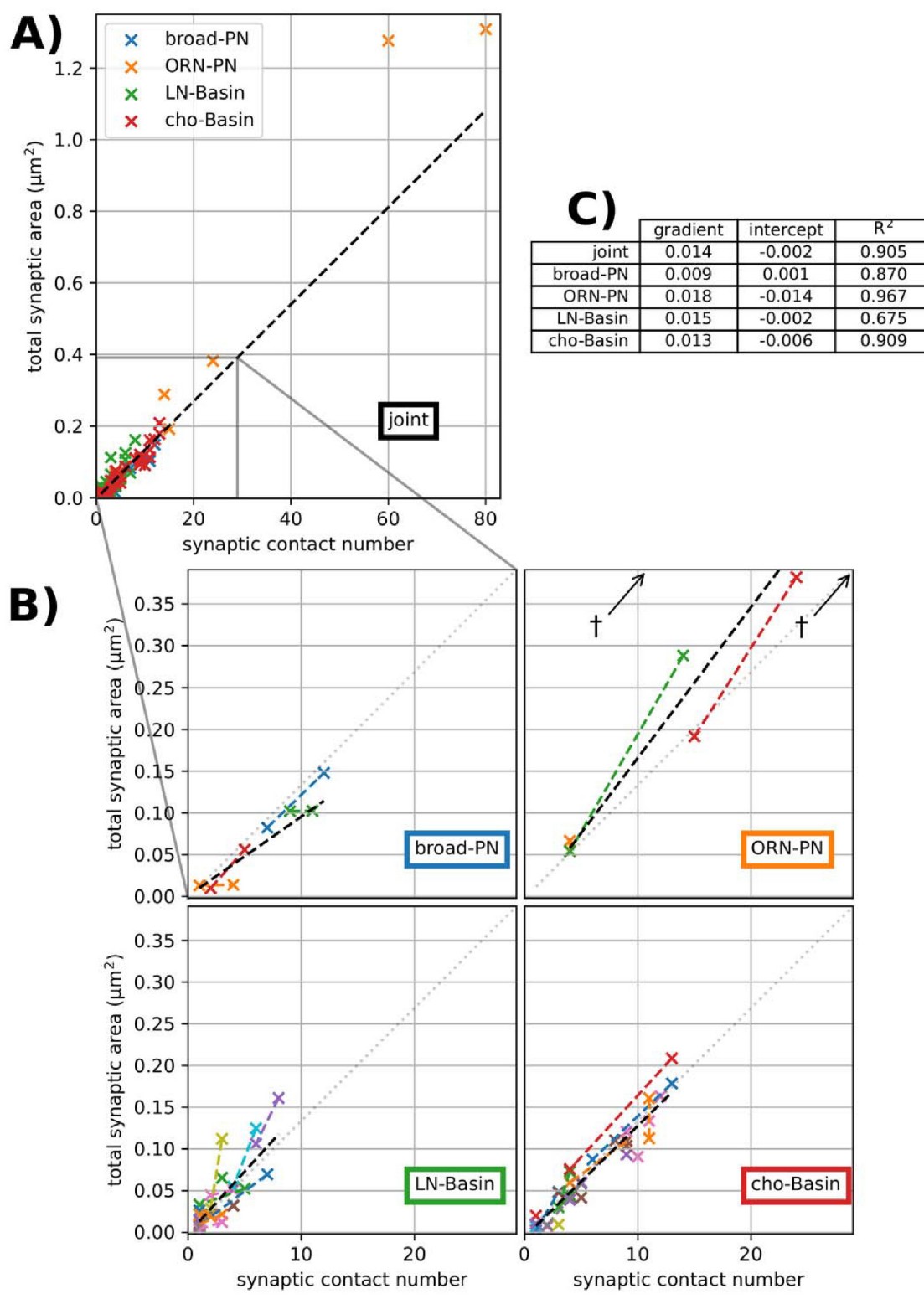

**Fig 3. Least-squares linear regressions of contact number vs area for each edge.** Weighted by the reciprocal of the contact number to reduce leverage by high-$n$ edges. **A)** Joint regression line for all edges (black dashed line), $y = 0.014x + 0.002$, $R^2 = 0.905$. **B)** For each circuit, a zoomed-in region of **A**, showing the joint regression (grey dotted line) and the circuit-specific regression line (black dashed line). Left-right pairs, when unambiguous, are shown in the same colour and joined with a dashed line of that colour. †shows outliers beyond the limits of the ORN-PN plot. **C)** Table of regression line gradient ($\mu m^2$count$^{-1}$ to 3 decimal places), y-intercept ($\mu m^2$ to 3 decimal places) coefficient of determination $R^2$ (to 3 decimal places).

It is assumed that the area of a single morphological synapse is the result of some stochastic process which depends on which edge it belongs to, and which type that edge is.

Here we enumerate the four levels of the hierarchical model:

$$A_{(i,j,k)} \mid (i,j) \sim \Gamma_{\text{shape}_0, \text{scale}_{0,(i,j)}}$$

**Level 0**
The areas $a$ of individual synaptic contacts $k$ within a single edge $(i,j)$ vary according to a gamma distribution whose scale depends on that edge's identity. There are 80 distributions $A$ described (one for each edge) with 540 samples observed (one for each morphological synapse).

$$\text{scale}_{0,(i,j)} \mid T_{(i,j)} = t \sim \Gamma_{\text{shape}_1, \text{scale}_{1,t}}$$

**Level 1**
The edge-specific scale parameters are gamma-distributed with a scale which depends on the edge type $T$. There are 4 distributions described, one for each edge type $t$.

$$\text{scale}_{1,t} \sim \Gamma_{\text{shape}_2, \text{scale}_2}$$

**Level 2**
The edge type-specific scale parameters are gamma-distributed with a scale hyperparameter.

$$\text{shape}_0, \text{shape}_1, \text{shape}_2, \text{scale}_2 \sim \text{U}$$

**Level 3**
Hyperparameters are sampled from a flat prior (i.e. a uniform distribution U).

This approach, with a hierarchical model (Fig 4A), gives us a single cohesive model which encodes the different sources of variance, the edge type being one of them. It allows us to use the same model when different amounts of information are available, and adjust our confidence in predicted values accordingly; a characteristic that makes the model well-suited for our data which has unequal sample sizes across edge types. The simple model (when we measure synaptic surface areas but ignoring the edge type or the partner identity) gives us a low-confidence prediction of the size of a newly-annotated synapse which we suspect to belong to this general distribution (Fig 4B). However, the prediction may change, and the confidence increase, if we know which edge type the synapse belongs to–finding whether this is true is the goal of this analysis. The confidence can increase again when we identify its presynaptic and postsynaptic neurons (the partner identity). In other words, within the Bayesian framework, the more prior information available, the better our prediction.

Furthermore, being able to parameterise the variance of the distribution parameters allows us to draw conclusions about how different the distributions are across edges and edge types. Tightly grouped $\text{scale}_0$ parameters within an edge type (low variance of $\Gamma_{\text{shape}_1, \text{scale}_{1,t}}$) would suggest that edges within that group have similar distributions. In this case, synaptic area measurements in one edge could be used to approximate the areas of synapses in other edges of the same type. A tight grouping of $\text{scale}_1$ parameters (low variance of $\Gamma_{\text{shape}_2, \text{scale}_2}$) would imply that the difference between edge types is small, and that synapse size information generalises across edge types.

Here we show how adding edge type information does not meaningfully improve the confidence in predicting total synapse area from contact number for an edge. Fig 4B shows the

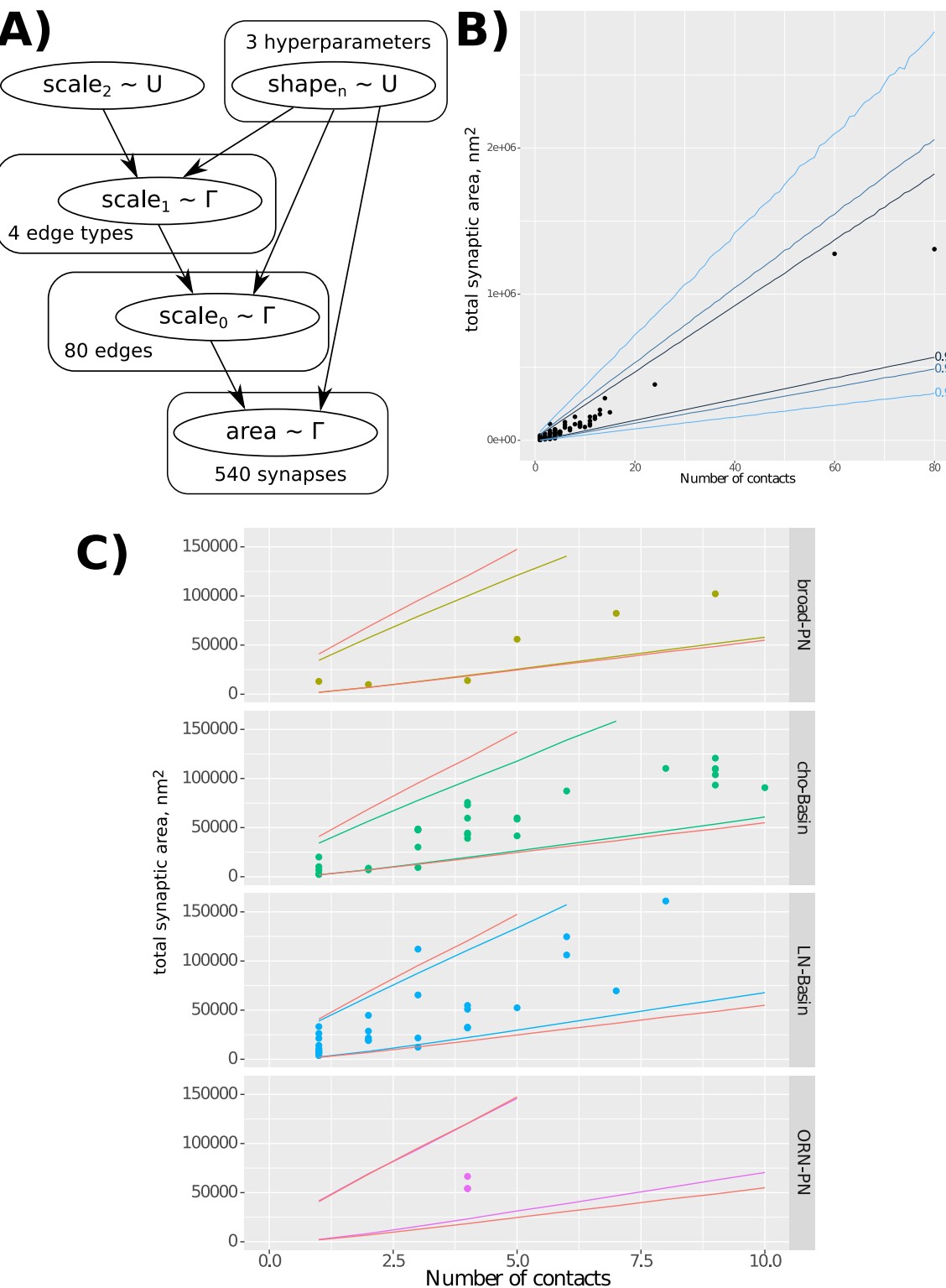

**Fig 4. Bayesian hierarchical modelling of distributions of synapse areas across graph edges and edge types. A)** Model dependency graph, showing how each level parametrises the levels below it, and the number of samples available at each level. **B)** Results of sampling from the generated model, including the bounds in which 90%, 95%, and 99% of samples fall, demonstrating that our data is in line with the model's expectation. **C)** Comparing the results of sampling sub-models with a fixed edge type with the full joint model (red lines): the small narrowing of the result intervals suggests that the edge type does not have predictive power of the relationship between contact number and contact area for a graph edge, and therefore that the sizes of individual synapses do not differ much between edge types.

confidence intervals for predicting total edge contact area given edge contact number, ignoring edge type information. Fig 4C shows, for each edge type, the change in confidence interval when information about that edge type is introduced to the prediction. The increase in prediction confidence (narrowing of the confidence interval) is small, suggesting synapse size information from one edge type generalises fairly well to other types, for the edge types studied here.

## Discussion

Our data shows that, within the studied connections in the *Drosophila* larval nervous system, the total synaptic surface area can be predicted solely from the number of morphological synaptic contacts from a presynaptic neuron to a postsynaptic one, independently of the participating neuronal cell types.

The diversity of edge types included in our study (sensory axons onto excitatory interneurons, and inhibitory interneurons onto excitatory interneurons), together with the observed strong correlation between contact counts and surface areas, suggest that we should expect a similarly strong correlation in other edge types in the central nervous system at the same *Drosophila* larval life stage.

Whether the observed correlation holds true through changes in neuronal arbor dimensions is an open question. In development, we know that the number of morphological synaptic contacts in a connection (i.e. an edge) grows by a factor of 5 from early first instar to late third, while the distribution of the number of partners per polyadic synapse remains constant [10], and the associated behavior doesn't change [35]. In addition, synaptic input fractions are preserved from first to third instar despite a 5-fold increase in cable and in number of morphological synaptic contacts [10]. Similarly, in the adult *Drosophila* fly, an increase of 50% in the amount of dendritic cable and number of morphological synaptic contacts doesn't alter the synaptic input fractions, or the density of synaptic contacts per unit of cable, or some key biophysical properties of the neuron [4]. Our data, together with these reports on the strong preservation of both structural and functional circuit properties across neuronal arbor growth, suggests that we ought to expect the observed strong correlation between number of contacts and surface areas to persist throughout larval development as well as in situations where arbors have different dimensions.

Our findings have implications for the study of the relationship between structure and function, in particular towards making functional inferences from morphological data. The estimation of synapse count from axo-dendritic apposition, also known as "Peters' rule" [36], has been shown not to hold in systems where it has been directly measured [37, 38]. On the other hand, synaptic area has a linear relationship with synaptic strength [14], so our findings (that synaptic count predicts synaptic area) suggest that inferring synaptic strength directly from the number of morphological contacts is grounded and generalisable across different edge types. Therefore our findings are consistent with the formulation of computational circuit models directly from morphological synaptic counts per edge, as has been done in previous studies that had implicitly assumed a correlation between counts and synaptic strength [2, 3, 7, 19, 39].

A full-fledged validation of our findings will require the comprehensive measurement of all synaptic surface areas of all neurons of a central nervous system. Recent and upcoming automated methods for segmenting synaptic surface areas [40], aided by improvements in volume EM such as isotropic imaging with FIBSEM [41], will make such a study tractable in the near term for small model organisms such as *Drosophila*.

Our findings have further implications towards speeding up reconstruction of wiring diagrams. Namely, comprehensive measurement of synaptic surface areas may no longer be a requirement. Therefore, the performance trade-off could be tipped towards imaging speed at the expense of resolution, lowering acquisition costs. As such, aiming for detecting synaptic puncta from light microscopy [9, 27, 29, 42] or low-resolution EM [41], would allow inference of synaptic strength, towards the computational analysis of neural circuits.

## Methods

### Imaging

The data volume used was the whole central nervous system described in [2]. The specimen was a 6 hour old female *Drosophila melanogaster*, in the L1 larval stage. It is comprised of 4850 sections, each 50*nm* thick, cut with a Diatome diamond knife. Each section was imaged at $3.8 \times 3.8nm$ resolution using an FEI Spirit TEM. The images were montaged and registered using the nonlinear elastic method described in [43].

### Neuronal and synapse morphologies

The neurons analyzed in this study were all published in [31] (olfactory sensory neurons ORNs, olfactory PNs and GABAergic Broad LNs) and in [3] (chordotonal somatosensory neurons, Basin neurons, and synaptically connected GABAergic LNs including Drunken, Ladder and Griddle LNs). Neuronal morphologies and connectivity were reconstructed collaboratively using CATMAID [44] with the procedures described in [38].

To reconstruct a neuronal arbor, point annotations ("skeleton nodes") are placed in each *z* section of a neurite. Synaptic contacts are identified by the presence of a thick black active zone, presynaptic specialisations such as vesicles and a T-bar, and evidence of postsynaptic specialisations, across several sections. In the CATMAID software, a point annotation ("connector node") is placed on the presynaptic side of the synaptic cleft; directed edges are then drawn from the nearest presynaptic skeleton node to the connector node, and from the connector node to a skeleton node in each postsynaptic neuron.

### Synaptic area annotation

Using catpy, a Python interface to CATMAID's REST API [45], the locations of synaptic contacts between neurons of interest were extracted, and an axis-aligned cuboid of image data including the connector node and the pre- and post-synaptic skeleton nodes of interest was downloaded. This was stored in an HDF5 [46] file whose schema extends that used in the CREMI challenge [47]. The location of the relevant skeleton and connector nodes are also stored in the file, to be used as a guide.

In *Drosophila*, most synapses are polyadic (one-to-many). Therefore, the best measurement for the synaptic contact area from presynaptic neuron *i* to postsynaptic neuron *j* is the area of the postsynaptic membrane. Using BigCAT, a BigDataViewer-based [48] volumetric annotation tool, the postsynaptic membrane (identified by evidence of postsynaptic specialisations, and adjacency to a synaptic cleft which was itself adjacent to presynaptic specialisations) was annotated with a thin line in each *z* section in which it appeared. These annotations are given unique IDs, and the association between the ID and the nodes it is associated with is also stored in the CREMI file.

These annotations were then normalised *post hoc*. Using scikit-image [49] v0.14, the line drawn in each *z* section was skeletonised [50]. The skeletonised line was converted into a string of coordinates in pixel space, which was smoothed with a Gaussian kernel ($\sigma = 3$). Its length in

pixel units was taken and multiplied by the *xy* resolution (3.8*nm*). This 2D length is then multiplied by the *z* resolution (50*nm*) to give an approximation of the synaptic surface represented by the membrane visible in this section, in $nm^2$. The area of a synapse is approximated by the sum of such areas for a single contact across *z* sections.

By extension, the contact area of a graph edge between neurons *i* and *j* is the sum of synaptic areas for all contacts between *i* and *j*.

### Hierarchical modelling

The gamma distribution was selected as it is a versatile continuous probability distribution with a low number of parameters and a positive domain. The priors for each of the 3 shape parameters were uniform between 0.001 and 1000. The prior for parameter scale$_2$ was uniform between 1 and 1000000.

The modeling was performed using the R language [51] and the JAGS [52] sampler. The sampler was run for 100000 iterations, and thinned by 2, with a 1000-iteration burn-in.

These scripts are included in the publication as a zip file, S1 File.

### Other software

The majority of analysis was performed using Python [53] version 3.7, numpy [54] v1.16, scipy [55] v1.1, and pandas [56] v0.23. Figs 1–3 were generated using matplotlib [57] v3.1 and FigureFirst [58]. Fig 4 was generated using the R language [51] and ggplot2 [59]. Figures were assembled using Inkscape [60] v0.94.

Data used for plotting are included in the publication as a zip file, S1 File.

### Supporting information

**S1 File. Hierarchical modelling scripts.** A compressed archive containing the Bayesian analysis code as an R package.
(ZIP)

**S2 File. Data tables used in plotting.** A compressed archive containing raw data used in plots as CSVs. Includes a README file with more information.
(ZIP)

### Acknowledgments

C.L.B. thanks the joint graduate program between HHMI Janelia Research Campus and the University of Cambridge, and William Schafer at the MRC LMB for being the Cambridge host; and thanks Susan Jones at the Department of Physiology, Development and Neuroscience of the University of Cambridge, and Maryrose Franko and Erik Snapp at HHMI Janelia, for their support during his graduate studies. The authors thank Andrew Champion and Tom Kazimiers for the development and support of CATMAID software, and Arlo Sheridan and Jan Funke at HHMI Janelia for their support with BigCAT software and the CREMI data.

### Author Contributions

**Conceptualization:** Christopher L. Barnes, Albert Cardona.

**Data curation:** Christopher L. Barnes.

**Formal analysis:** Christopher L. Barnes, Daniel Bonnéry.

**Funding acquisition:** Albert Cardona.

**Methodology:** Christopher L. Barnes, Daniel Bonnéry, Albert Cardona.

**Project administration:** Albert Cardona.

**Resources:** Albert Cardona.

**Software:** Christopher L. Barnes, Daniel Bonnéry, Albert Cardona.

**Supervision:** Daniel Bonnéry, Albert Cardona.

**Validation:** Albert Cardona.

**Visualization:** Christopher L. Barnes, Albert Cardona.

**Writing – original draft:** Christopher L. Barnes, Albert Cardona.

**Writing – review & editing:** Christopher L. Barnes, Albert Cardona.

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
