## [Decision Letter · Decision Letter 0]

25 Jan 2022

PONE-D-21-37427Synaptic counts approximate synaptic contact area in DrosophilaPLOS ONE

Dear Dr. Cardona,

We are providing some comments that hopefully will help you improve the manuscript, in particular regarding its position within the existing literature in the field. We suggest the work will gain greater impact and reach a greater audience if its conclusion were to be discussed more generally within the neuroscience realm.

Some of the overarching conclusion may also benefit from a more robust review of existing literature.

We look forward to receiving your revised manuscript.

Kind regards,

Giorgio F Gilestro, PhD

Academic Editor

PLOS ONE

Journal Requirements:

4. Please update your submission to use the PLOS LaTeX template. The template and more information on our requirements for LaTeX submissions can be found at http://journals.plos.org/plosone/s/latex.

Reviewers' comments:

Reviewer's Responses to Questions

**Comments to the Author**

1. Is the manuscript technically sound, and do the data support the conclusions?

Partly

2. Has the statistical analysis been performed appropriately and rigorously?

Yes

3. Have the authors made all data underlying the findings in their manuscript fully available?

Yes

4. Is the manuscript presented in an intelligible fashion and written in standard English?

Yes

5. Review Comments to the Author

Reviewer #1: Barnes et al demonstrate that that excitatory and inhibitory inputs onto Olfactory projections neurons (PNs) and first-order mechano/nociceptive neurons (Basins) exhibit a very simple relationship between the number of contacts and the total surface area of the synaptic surface area. The data is rigorously analysed, well presented and the conclusions drawn from this work are clearly stated. In their conclusions, the authors argue that simply measuring the number of synaptic contacts between neuron A and B could be used as a surrogate marker of synaptic strength. If true, this would greatly simplify some of the challenges associated with connectome analysis.

I was surprised that no mention was made of Peters rule – a term that was applied by others to the work of Peters and Feldman (J Neurocytol. 1976 Feb; 5(1):63-84) – that close apposition of axons to dendrites can be used to predict the number of synapses. This simple rule is still widely debated and not at all proven. The data presented by Barnes et al builds on Peters rule as they show that the number of contacts would correlate well with synaptic area.

However, my main concern with this manuscript is that there is a lack of direct evidence cited in the current manuscript to support the view that synaptic area is a measure of synaptic strength. Barnes et al rely on three very important research papers to support this key argument. Unfortunately, I do not see how any of these papers support this idea. The seminal work of Castillo & Katz in the 1950’s is somehow used to make the claim that synaptic strength correlates with release probability. I cannot see how this claim can be made from the elegant studies on the quantal nature of the miniature end plate potential by Castillo & Katz. The authors then cite Branco et al (2010) to claim that vesicle release probability correlates with the number of docked vesicles. However, the paper by Branco et al (2010) reached entirely the opposite conclusion about synaptic area and release probability. This paper clearly states that although there is a positive correlation between release probability and the number of docked vesicles the synaptic area does not correlate with the release probability. The final paper (Ikeda & Bekkers, 2009) does not contain any anatomical data concerning docked vesicles and uses a purely functional approach based upon blocking transmitter recycling to estimate the reserve pool of vesicles and I do not see the relevance of this paper to the arguments made by Barnes et al. Therefore, I feel that this aspect of the conclusion needs a lot more work.

There are many other studies that have attempted to address the question of whether synaptic area correlates with synaptic strength, but these studies were overlooked in the current manuscript. In particular, I am reminded of the work of Farrant, Cull-Candy & Nusser (1997). This study combined whole-cell recording with quantitative immunolabelling at EM resolution to conclude that variation in receptor number at the synapse largely explains variability in mIPSC amplitude. Importantly for Barnes et al, receptor density appears uniform and so surface area could be used to predict receptor number. Unfortunately, receptor density may not be uniform at all synapse types and I am not aware this parameter is known for Drosophila – it certainly is not reported in the current manuscript. In the absence of this data or any functional data on the synaptic strength recorded at the excitatory and inhibitory synapses onto PN and Basin neurons of Drosophila I am concerned that the importance of this work is over-stated. However, the analysis performed on the data-sets in this study are impressive and I am sure this work will be of great value to many involved in connectome research.

6. PLOS authors have the option to publish the peer review history of their article (what does this mean?). If published, this will include your full peer review and any attached files.

Do you want your identity to be public for this peer review? For information about this choice, including consent withdrawal, please see our Privacy Policy.

**No**

---

## [Author Response · Author response to Decision Letter 0]

10 Mar 2022

Thank you for the helpful comments. We have now revised the text to address the reviewer's comments. Please see attached PDF file named "Response to Reviewers.pdf" for a point by point response.

---

## [Editor Report · Decision Letter 1]

14 Mar 2022

Synaptic counts approximate synaptic contact area in Drosophila

PONE-D-21-37427R1

Dear Dr. Cardona,

We’re pleased to inform you that your manuscript has been judged scientifically suitable for publication and will be formally accepted for publication once it meets all outstanding technical requirements.

Kind regards,

Giorgio F Gilestro, PhD

Academic Editor

PLOS ONE
---

## [Editor Report · Acceptance letter]

25 Mar 2022

PONE-D-21-37427R1 

Synaptic counts approximate synaptic contact area in *Drosophila*

Dear Dr. Cardona:

I'm pleased to inform you that your manuscript has been deemed suitable for publication in PLOS ONE. Congratulations! Your manuscript is now with our production department. 

Kind regards, 

on behalf of

Dr. Giorgio F Gilestro 

Academic Editor

PLOS ONE